# Correcting the hebbian mistake: Toward a fully error-driven hippocampus

**Yicong Zheng**[1,2], **Xiaonan L. Liu**[3], **Satoru Nishiyama**[4,5], **Charan Ranganath**[1,2], **Randall C. O'Reilly**[1,2,6] *

1 Department of Psychology, University of California, Davis, California, United States of America, 2 Center for Neuroscience, University of California, Davis, California, United States of America, 3 Department of Psychology, The Chinese University of Hong Kong, Hong Kong, People's Republic of China, 4 Graduate School of Education, Kyoto University, Kyoto, Japan, 5 Japan Society for the Promotion of Science, Tokyo, Japan, 6 Department of Computer Science, University of California, Davis, California, United States of America

* oreilly@ucdavis.edu

**Data Availability Statement:** Full simulation code and relevant data is available at: https://github.com/ccnlab/hip-edl.

**Funding:** RCO, CR were funded by the Office of Naval Research, grant numbers: N00014-20-1-

## Abstract

The hippocampus plays a critical role in the rapid learning of new episodic memories. Many computational models propose that the hippocampus is an autoassociator that relies on Hebbian learning (i.e., "cells that fire together, wire together"). However, Hebbian learning is computationally suboptimal as it does not learn in a way that is driven toward, and limited by, the objective of achieving effective retrieval. Thus, Hebbian learning results in more interference and a lower overall capacity. Our previous computational models have utilized a powerful, biologically plausible form of error-driven learning in hippocampal CA1 and entorhinal cortex (EC) (functioning as a sparse autoencoder) by contrasting local activity states at different phases in the theta cycle. Based on specific neural data and a recent abstract computational model, we propose a new model called Theremin (Total Hippocampal ERror MINimization) that extends error-driven learning to area CA3—the mnemonic heart of the hippocampal system. In the model, CA3 responds to the EC monosynaptic input prior to the EC disynaptic input through dentate gyrus (DG), giving rise to a temporal difference between these two activation states, which drives error-driven learning in the EC→CA3 and CA3↔CA3 projections. In effect, DG serves as a teacher to CA3, correcting its patterns into more pattern-separated ones, thereby reducing interference. Results showed that Theremin, compared with our original Hebbian-based model, has significantly increased capacity and learning speed. The model makes several novel predictions that can be tested in future studies.

## Author summary

Exemplified by the famous case of patient H.M. (Henry Molaison) whose hippocampus was surgically removed, the hippocampus is critical for learning and remembering everyday events—what is typically called "episodic memory." The dominant theory for how it learns is based on the intuitive principle stated by Donald Hebb in 1949, that neurons that

2578. RCO was funded by the Office of Naval
Research, grant numbers: N00014-19-1-2684,
N00014-18-C-2067. The funders had no role in
study design, data collection and analysis, decision
to publish, or preparation of the manuscript.

**Competing interests:** I have read the journal's
policy and the authors of this manuscript have the
following competing interests: R. C. O'Reilly is
Director of Science at Obelisk Lab in the Astera
Institute, and Chief Scientist at eCortex, Inc., which
may derive indirect benefit from the work
presented here.

"fire together, wire together"—when two neurons are active at the same time, the strength of their connection increases. We show in this paper that using a different form of learning based on correcting errors (error-driven learning) results in significantly improved episodic memory function in a biologically-based computational model of the hippocampus. This model also provides a significantly better account of behavioral data on the testing effect, where learning by testing with partial cues is better than learning with the complete set of information.

## Introduction

It is well-established that the hippocampus plays a critical role in the rapid learning of new episodic memories [1]. Most computational and conceptual models of this hippocampal function are based on principles first articulated by Donald O. Hebb and David Marr [2–5]. At the core of this framework is the notion that recurrent connections among CA3 neurons are strengthened when they are co-activated ("cells that fire together, wire together"), essentially creating a cell assembly of interconnected neurons that bind the different elements of an event. As a result of this Hebbian learning, subsequent partial cues can drive pattern completion to recall the entire original memory, by reactivating the entire cell assembly via the strengthened interconnections.

In addition, Marr's fundamental insight was that sparse levels of neural activity in area CA3 and especially the dentate gyrus (DG) granule cells, will drive the creation of cell assemblies that involve a distinct combination of neurons for each event, otherwise known as *pattern separation* [3, 6, 7]. As a consequence, the DG to CA3 pathway has the capability to minimize interference from learning across even closely overlapping episodes (e.g., where you parked your car today vs. where you parked it yesterday). Note that it is the patterns of activity over area CA3 that constitute the principal hippocampal representation of an episodic memory, and learning in these CA3 synapses is thus essential for cementing the storage of these memories. Overall, the basic tenets established by Hebb and Marr account for a vast amount of behavioral and neural data on hippocampal function, and represents one of the most widely accepted theories in neuroscience [8–11]

Although almost every biologically-based computational model of hippocampal function incorporates Hebbian plasticity, it is notable that Hebbian learning is computationally suboptimal in various respects, especially in terms of overall learning capacity [12, 13]. In models that rely solely on Hebbian learning, the local activity of the sending and receiving neurons drives synaptic weight changes, regardless of how necessary those changes might be to achieve better memory recall. As a result, such models do not know when to stop learning, and continue to drive synaptic changes beyond what is actually necessary to achieve effective pattern completion. The consequence of this "learning overkill" is that all those unnecessary synaptic weight changes end up driving more interference with the weights needed to recall other memories, significantly reducing overall memory capacity. Even the high degree of pattern separation in the DG and CA3 pathways might not be sufficient to make up for the interference caused by reliance on Hebbian learning. Although it is difficult to quantitatively assess the capacity of the hippocampus in various species, there is reason to believe that even the high degree of pattern separation in the DG and CA3 pathways might not be sufficient to make up for the interference caused by reliance on Hebbian learning.

Although the simplest form of Hebbian learning is widely understood to be impractical given that weights are unbounded, the various forms of normalization and bounding that are

widely used (which we had incorporated into our previous hippocampal models), including BCM [14] and Oja's rule and variants [15, 16], still do not have a way of limiting learning in relation to overall recall success. Thus, a logical alternative to the Hebbian approach is to introduce a self-limiting learning mechanism that drives only synaptic changes that are absolutely necessary to support effective function. However, determining this minimal amount of learning can be challenging: how can local synaptic changes "know" what is functionally necessary in terms of the overall memory system function? One well-established class of such learning mechanisms are error-driven learning rules: by driving synaptic changes directly in proportion to a functionally-defined error signal, learning automatically stops when that error signal goes to zero. For example, the well-known Rescorla-Wagner learning rule for classical conditioning [17] is an instance of the delta-rule error-driven learning rule [18]:

$$dW = x(r - y),\tag{1}$$

where $dW$ is the amount of synaptic weight change, $x$ is the sending neuron activity level (e.g., average firing rate of sensory inputs representing conditioned stimuli), $r$ is the actual amount of reward received, and $y$ is the expected amount of reward, computed according to the existing synaptic weights:

$$y = xW,\tag{2}$$

This learning rule drives learning (changes in weights, $dW$) up to the point where the expected prediction of reward ($y$) matches the actual reward received ($r$), at which point learning stops, because the difference term in the parentheses goes to 0. The dependency on $x$ is critical for *credit assignment*, which ensures that the most active sending neurons change their weights the most, as such weight changes will be the most effective in reducing the error. The widely-used error backpropagation learning algorithm is a mathematical extension of this simpler delta-rule form of learning [19], and demonstrates the general-purpose power of these error-driven learning principles, underlying the current success in large-scale deep learning models [20].

We have previously shown that these error-driven learning principles can be applied to the CA1 region of the hippocampus [21], building on phase-dependent CA1 patterns (EC-dominated encoding mode at theta troughs vs. CA3-dominated retrieving mode at theta peaks) discovered by [22]. The critical *target* value driving this error-driven learning is the full pattern of activity over the entorhinal cortex (EC) representing the current state of the rest of the cortex. Learning stops when the hippocampal encoding of this EC pattern projecting from CA3 through CA1 matches the target version driven by the excitatory projections into the EC. These error-driven learning dynamics have been indirectly supported empirically by studies of CA1 learning in various tasks [23, 24]. However, this prior model retained the Hebbian learning (in a bounded form related to Oja's rule, known as conditional principal components analysis; CPCA; [16]) for all of the connections within CA3 and DG, because the error signal that drives CA1 learning does not have any way of propagating back to these earlier areas within the overall circuit: the connectivity is only feedforward from CA3 to CA1.

To be able to apply a similar type of self-limiting error-driven learning to the core area CA3 of the hippocampus, we need a suitable target signal available to neurons within the CA3 that determines when the learning has accomplished what it needs to do. Recently, [25] proposed in an abstract, backpropagation-based model that the DG can serve as a kind of teacher to the CA3, driving learning just to the point where CA3 on its own can replicate the same highly pattern-separated representations that the DG imparts on the CA3. We build on this idea here, by showing how error-driven learning based on this DG-driven target signal can emerge

naturally within the activation dynamics of the hippocampal circuitry, driving learning in the feedforward and recurrent synapses of area CA3. Thus, we are able to extend the application of error-driven learning to the "heart" of the hippocampus.

By adding the CA3 error-driven learning mechanism, we show that this more fully error-driven hippocampal learning system has significantly improved memory capacity and resistance to interference compared to one with Hebbian learning in CA3. Furthermore, we show how these error-driven learning dynamics fit with detailed features of the neuroanatomy and physiology of the hippocampal circuits, help us understand the nature of memory encoding and retrieval, and can have broad implications for understanding important learning phenomena such as the *testing effect* [26]. Overall, this new framework provides a coherent computational and biological account of hippocampal episodic learning, departing from the tradition of Hebbian learning at a computational level, while retaining the overall conceptual understanding of the essential role of the hippocampus in episodic memory. Thus, we do not throw the baby out with the bathwater here, and our model still matches the vast majority of behavioral and neural data consistent with the classic Hebb-Marr model. Nevertheless, it also does make novel predictions, and at a broad, behavioral level, the improved performance of our model is more compatible with the remarkable capacity of the relatively small hippocampal system for encoding so many distinct memories over the course of our lives.

In the remainder of the paper, we first introduce the computational and biological framework for error-driven learning in the hippocampal circuits, and then present the details of an implemented computational model, followed by results of this model as compared to our previous Hebbian-CA3 version [21], as well as representational analyses that capture the subregional dynamics in the model. We also present how the testing effect might arise due to the error-driven dynamics implemented in our model, compared to the same model but without CA3 error-driven learning. We conclude with a general discussion, including testable predictions from this framework and implications for some salient existing behavioral and neural data on hippocampal learning.

## Sources of error driven learning in the hippocampal circuit

We begin by briefly reviewing our earlier work showing how the monosynaptic pathway interconnecting the EC and CA1 can support error-driven learning, via systematic changes in pathway strengths across the theta cycle [21, 22, 27] (Fig 1). Although nominally a central part of the hippocampus, from a computational perspective it makes more sense to think of this monosynaptic CA1 ↔ EC pathway (sometimes known as the temporo-ammonic pathway) as an extension of the neocortex, where the principles of error-driven learning have been well-developed [28–30]. Specifically, this pathway can be thought of as learning to encode the EC information in CA1 in a way that can then support reactivation of the corresponding EC activity patterns when memories are later retrieved via CA3 pattern completion. Computationally, this is known as an *auto-encoder*, and error-driven learning in this case amounts to adjusting the synapses in this monosynaptic pathway to ensure that the EC pattern is accurately reconstructed from the CA1 activity pattern.

The differential modulation of the pathway strengths as shown in Fig 1 is the primary driver of error-driven learning in the monosynaptic pathway, based on the critical data from [22]. Starting with the pathway from CA1 to ECout, the weak-then-strong drive from ECin to ECout provides the opportunity for ECout to exhibit two different states of activity: first the state of activity where it is primarily reflecting the CA1 → ECout projection, and then, in the final *plus phase* or *Fourth Quarter* of the theta cycle, the state where ECout reflects the driving input from ECin. In terms of the delta rule equation shown above, these two states of ECout

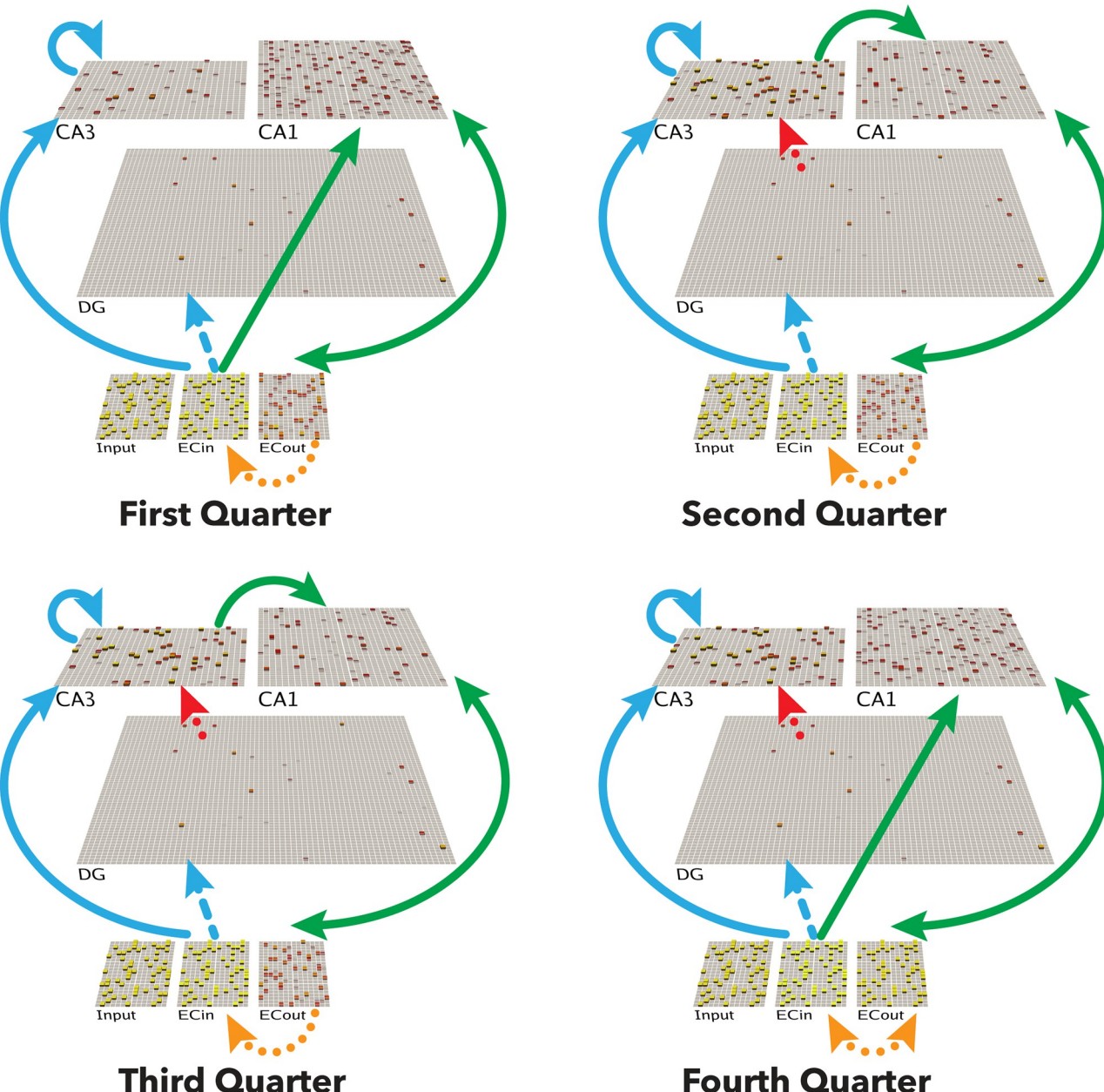

**Fig 1. Architecture of the Theremin model.** Visual depiction of one full theta-cycle training trial, separated into four different phases within the cycle (i.e., four *quarters*, each representing 50 ms). The CA1 learns to properly decode the CA3 pattern into the corresponding EC representation, while CA3 learns to encode the EC input in a more pattern-separated manner reflecting DG input. Arrows depict pathways of particular relevance for that quarter. **First Quarter**: Blue arrows show initial activation of CA3 and DG via monosynaptic pathways from ECin (superficial layers of EC). Green arrows show CA1 likewise being monosynaptically driven from ECin, and in turn driving ECout (deep layers) with bidirectional connectivity. **Second Quarter**: Red arrow indicates DG driving CA3, providing a target activity state over CA3 relative to the first quarter state. Also, CA3 starts to drive CA1, resulting in full "attempted recall" state over ECout by the end of the **Third Quarter**. **Fourth Quarter**: the ECin drives ECout (Orange arrow), which in turn drives any resulting changes in CA1. Note: The fourth quarter is the plus phase for all error-driven learning projections, the second quarter and the third quarter are the minus phase for CA3 → CA1, and the first quarter is the minus phase for ECin → CA1, CA3 → CA3, ECin → CA1, and CA1 ↔ ECout (see Methods for more details). Solid lines represent projections that have error-driven learning + Hebbian learning, dashed lines represent projections that only have Hebbian learning, dotted lines represent projections that do not learn in the model.

enable error driven learning as follows:

$$dW_{i,j} = CA1_i(ECout_j^{ECin} - ECout_j^{CA1}), \tag{3}$$

where $ECout_j^{CA1}$ represents the activity of the neuron $j$ in the ECout layer driven more strongly by its CA1 afferents (earlier in the theta cycle), and $ECout_j^{ECin}$ represents the state when driven more strongly by the ECin afferents, at the end of the theta cycle (Fig 1). If these activities are identical, then the error is 0, and no learning occurs (i.e., $dW_{i,j} = 0$), and any learning that does occur is directly in proportion to the extent of error correction required, to get the CA1 to ECout pathway to more accurately reproduce the content of the ECin information. Biologically, there are relatively focal "columnar" projections from the superficial (ECin) to deep (ECout) layers of EC, consistent with cortical anatomical organization in general [31–33], and in our model, we just use direct fixed one-to-one connections so that ECout literally mirrors the organization of ECin, but any information preserving connectivity pathway here would function similarly.

This error-driven learning mechanism also applies to the pathways of ECin → CA1 and ECout → CA1, by virtue of the differential influence of the ECin, ECout → CA1 pathways on activity of the CA1 neurons across the theta cycle. For example, CA1 receives from both the CA3 and the EC layers, and the differential strength of these pathways creates different CA1 activity states across the theta cycle, similar to Eq (3):

$$dW_{i,j} = CA3_i(CA1_j^{ECin} - CA1_j^{CA3}), \tag{4}$$

where $CA1_j^{CA3}$ reflects CA1 activity when driven more strongly by the CA3 inputs earlier in the theta cycle, while $CA1_j^{ECin}$ reflects the final activity state when driven more strongly by ECin (and also by the bidirectional ECout pathway, where ECout is likewise being more strongly driven by ECin). The ability of these phasic differences in activity state to reverberate across layers in bidirectionally-connected networks, and thus drive error-driven learning even in further-away areas, produces a close mathematical approximation to the error backpropagation algorithm [28–30]. This general form of error-driven learning converges on the same temporal-difference *contrastive hebbian learning* (CHL) formulation as the original Boltzmann machine [34], where the target phase is called the *plus phase*, from which the prior *minus phase* is subtracted [16, 35, 36]. See [21] for more details on learning in this EC ↔ CA1 monosynaptic pathway. Consistent with the idea that this monosynaptic pathway is more cortex-like in nature, [24] have shown that this pathway can learn to integrate across multiple learning experiences to encode sequential structure, in a way that depends critically on the error-driven nature of this pathway, and is compatible with multiple sources of data [23].

To extend this error-driven learning mechanism to area CA3, which is the primary objective of this paper, it is essential to have two different activation states, one that represents a *target* representation (e.g., the actual reward, or the actual ECin input in the examples considered previously), and the other that represents what the current synaptic weights produce on their own. The key idea in this new Theremin model is that the highly pattern-separated activity pattern in the DG drives a target representation as a pattern of activity over CA3, which drives error-driven learning relative to the activity state of CA3 based on the direct ECin → CA3 projections (i.e., prior to the arrival of DG → CA3 inputs). Thus, in effect, the DG, which is the sparsest and most pattern-separated hippocampal layer, is serving as a teacher to the CA3, driving error-driven learning signals there just to the point where CA3 on its own can replicate the DG-driven sparse, pattern-separated representations [25].

The delta-rule formulation for this new error-driven learning component is:

$$dW_{i,j} = \text{ECin}_i(\text{CA3}_j^{DG} - \text{CA3}_j^{ECin}), \tag{5}$$

where $\text{CA3}_j^{ECin}$ is the activity of the CA3 neuron $j$ prior to the arrival of the DG input, based on the $\text{ECin}_i$ inputs, and $\text{CA3}_j^{DG}$ is the activity of CA3 after the DG inputs arrive, which is at least 5ms later as we review below. Thus, as in the above cases, the temporal difference between CA3 patterns can drive synaptic weight changes $dW_{i,j}$, representing the change in synaptic weight between $\text{ECin}_i$ and $\text{CA3}_j$. Critically, to the extent that CA3 prior to DG input is already matching the DG-driven pattern, no additional learning needs to occur, thus producing the interference minimization benefits of error-driven learning. Note that the same error-driven signal in CA3 trains the lateral recurrent pathway within CA3 in addition to the ECin → CA3 perforant pathway (PP) projections (Fig 1), so that these recurrent connections also adapt to fit the DG-driven pattern, but no further.

Although this form of error-driven learning might make sense computationally, how could something like this delta error signal emerge naturally from the hippocampal biology? First, as in our prior model of learning in the monosynaptic pathway [21], this error signal emerges naturally as a *temporal difference* between two states of activity over the CA3, which is also consistent with a broader understanding of how error-driven learning works in the neocortex [16, 28, 36]. Specifically, the appropriate temporal difference over CA3 arises from the additional delay associated with the propagation of the MF signal through the DG to the CA3, compared to the more direct PP signal from ECin to CA3. Thus, the *minus phase* term in the delta rule occurs first (i.e., $\text{CA3}^{ECin}$), followed by the *plus phase* (i.e., $\text{CA3}^{DG}$)—this terminology goes back to the Boltzmann machine, which also used a temporal-difference error-driven learning mechanism; [34].

Second, this error-driven learning could arise from *heterosynaptic plasticity* in the hippocampus, meaning that activation from one type of sending neuron (i.e., the DG in this case) leads to synaptic changes in other input synapses (i.e., from EC) to the same receiving neuron (CA3). Specifically, activation of the strong, anatomically unique mossy fiber inputs from DG could drive plasticity in the PP and CA3 recurrent connections, in contrast to the more typical homosynaptic plasticity case where activity local to the synapse drives its plasticity. Neurophysiologically, there are a number of lines of empirical evidence consistent with these potential mechanisms:

- CA3 pyramidal cells respond to PP stimulation prior to the granule cells in the DG, in vivo [37, 38], such that the indirect input through the DG will be delayed due to the slower DG response (by roughly 5 msec at least).

- MF inputs from the DG granule cells to the CA3 pyramidal cells can induce heterosynaptic plasticity at PP and CA3 recurrent connections [39–42]. This is consistent with ability of the later-arriving DG inputs to drive the CA3 synaptic changes toward that imposed by this stronger target-like pattern, compared to the earlier pattern initially evoked by PP and CA3 recurrent inputs.

- Although several studies have found that contextual fear learning is intact without MF input to CA3 [43–45], incomplete patterns from DG during encoding impair the function of EC → CA3 pathway in contextual fear conditioning tasks [46], suggesting that DG still plays an important role in heterosynaptic plasticity at CA3.

In addition to this DG-driven error learning in CA3, we explored a few other important principles that also help improve overall learning performance. First, reducing the strength of

the MF inputs to the CA3 during memory recall helped shift the dynamics toward pattern completion instead of pattern separation, as was hypothesized in [6]. This is consistent with evidence and models showing that MF projections are not necessary in naturally recalling a memory [44, 46, 47]. However, other data suggests that it still plays an important role in increasing recall precision [44, 46, 48, 49]. Thus, consistent with these data, we found that reducing, but not entirely eliminating MF input to the CA3 during recall was beneficial, most likely because it enabled the other pathways to exert a somewhat stronger influence in favor of pattern completion, while still preserving the informative inputs from the DG.

Second, we experimented with the parameters on the one remaining Hebbian form of learning in the network, which is in the ECin → DG pathway (i.e., PP). This pathway does not have an obvious source of error-driven contrast, given that there is only one set of projections into the DG granule cells. Thus, we sought to determine if there were particular parameterizations of Hebbian learning that would optimize learning in this pathway, and found that shifting the balance of weight decreases over weight increases helped learning overall, working to increase pattern separation in this pathway still further.

Finally, we tested a range of different learning rates for all of the pathways in the model, along with relative strengths of the projections, across a wide range of network sizes and numbers of training items, to determine the overall best parameterization under these new mechanisms.

Next, we describe our computational implementation within the existing [21] framework, and then present the results of a systematic large-scale parameter search of all relevant parameters in the model, to determine the overall best-performing configuration of the new model.

## Methods

### Hippocampal architecture

The current model, which we refer to as the Theremin (i.e., Total Hippocampal ERror MINimization) (Fig 1), is based on our previous theta-phase hippocampus model [21], which was developed within the earlier Complementary Learning System (CLS) model of the hippocampus [50, 51]. The broader implementation framework is the Leabra model (Local, Error-driven, and Associative, Biologically Realistic Algorithm), which provides point-neuron rate-coded neurons, inhibitory interneuron-mediated competition and sparse, distributed representations, full bidirectional connectivity, and temporal-difference based error-driven learning dynamics [16, 35] in the Emergent software. See https://github.com/emer/leabra for fully-documented equations, code, and several example simulations, including the exact model presented here. The S1 Appendix also contains a summary of the key mechanisms and equations.

Fig 1 shows the hippocampal architecture captured in our models. The EC superficial layer (ECin) is the source of input to the hippocampus, integrated from all over the cortex. Based on anatomical and physiological data, we organize the EC into different pools (also called slots) that reflect the inputs from different cortical areas, and thus have different types of representations reflecting the specializations of these different areas [32]. In the present model, we assume some pools reflect item-specific information, while others reflect the various aspects of information that together constitute context, which is important for distinguishing different memory lists in our tests.

The ECin projects to the DG and CA3 via broad, diffuse PP projections, which have a uniform 25% random chance of connection. This connectivity is essential for driving conjunctive encoding in the DG and CA3, such that each receiving neuron receives a random sample of

information across the full spectrum present in the ECin. Further, the DG and CA3 have high levels of inhibition, driving extreme competition, such that only those neurons that have a particularly favorable conjunction of input features are able to get active in the face of the strong inhibition. This is the core principle behind Marr's pattern separation mechanism, captured by his simple R-theta codon model [3]. Using the FFFB (feedforward & feedback) inhibition mechanism in Leabra, DG has a inhibitory conductance multiplier of 3.8 and CA3 has 2.8, compared to the default cortical value of 1.8 which produces activity levels of around 15%. These resulted in DG activity around 1% and CA3 around 2%. The number of units in DG is roughly 5 times of that in CA3, consistent with the theta-phase hippocampus model.

The CA3 receives strong MF projections from the DG, which have a strength multiplier of 4 (during encoding), giving the DG a much stronger influence on CA3 activity compared to the direct PP inputs from ECin. CA3 also receives recurrent collateral projections which have a strength multiplier of 2, which are the critical Hebbian cell-assembly autoassociation projections in the standard Hebb-Marr model, as captured in [21] using a Hebbian learning mechanism. That model also uses Hebbian learning in the PP pathways from ECin to DG and CA3, which also facilitate pattern completion during recall as analyzed in [6].

In the monosynaptic pathway, ECin (superficial) layers project to CA1, which then projects back into the deep layers of EC, called ECout in the model, such that CA1 encodes the information in ECin and can drive ECout during recall to drive the hippocampal memory back out into the cortex. This is the auto-encoder function of CA1, which is essential for translating the highly pattern-separated representations in CA3 back into the "language" of the cortex. Thus, a critical locus of memory encoding is in the CA3 → CA1 connections that associate the CA3 conjunctive memory with the CA1 decoding thereof—without this, the randomized CA3 patterns would be effectively unintelligible to the cortex.

Unlike the broad and diffuse PP projections, the EC ↔ CA1 connections obey the pool-wise organization of EC, consistent with the focal, point-to-point nature of the EC ↔ CA1 projections [32]. Thus, each pool is separately auto-encoding the specific, specialized information associated with a given cortical area, which enables these connections to slowly learn the systematic "language" of that area. The entire episodic memory is thus the distributed pattern across all of these pools, but the monosynaptic pathway only sees a systematic subset, which it can efficiently and systematically auto-encode.

The purpose of the theta-phase error-driven learning in the [21] model is to shape these synaptic weights to support this systematic auto-encoding of information within the monosynaptic pathway. Specifically, CA1 patterns at peaks and troughs of theta cycles come from CA3-retrieved memory and ECin inputs, respectively. As shown in Eq (3) above, the target plus-phase activation comes from the ECout being strongly driven by the ECin superficial patterns, in contrast to the minus phase where ECout is being driven directly by CA1. Thus, over iterations of learning, this error-driven mechanism shapes synapses so that the CA1 projection to ECout will replicate the corresponding pattern on ECin.

Relative to this theta-phase model, the new Theremin model introduces error-driven learning in the CA3, using Eq (5) as shown above, which was achieved by delaying the output of DG → CA3 until the second quarter (Fig 1). Although it only takes ∼10 ms for signals to propagate from ECin to DG to CA3 in the rodent hippocampus, and ∼5 ms from ECin to CA3 monosynaptically, fully activated CA3 patterns do not show up until ∼50 ms (equivalent to a quarter of a 200 ms theta cycle as modeled here; [52]). Thus, the plus phase (the fourth quarter) represents the CA3 activity in the presence of these strong DG inputs, while the minus phase (the first quarter) is the activity prior to the activation of these inputs. In addition, as noted earlier, we tested the effects of reducing the strength of MF inputs to CA3 during recall testing, along with testing all other relevant parameters in a massive grid search.

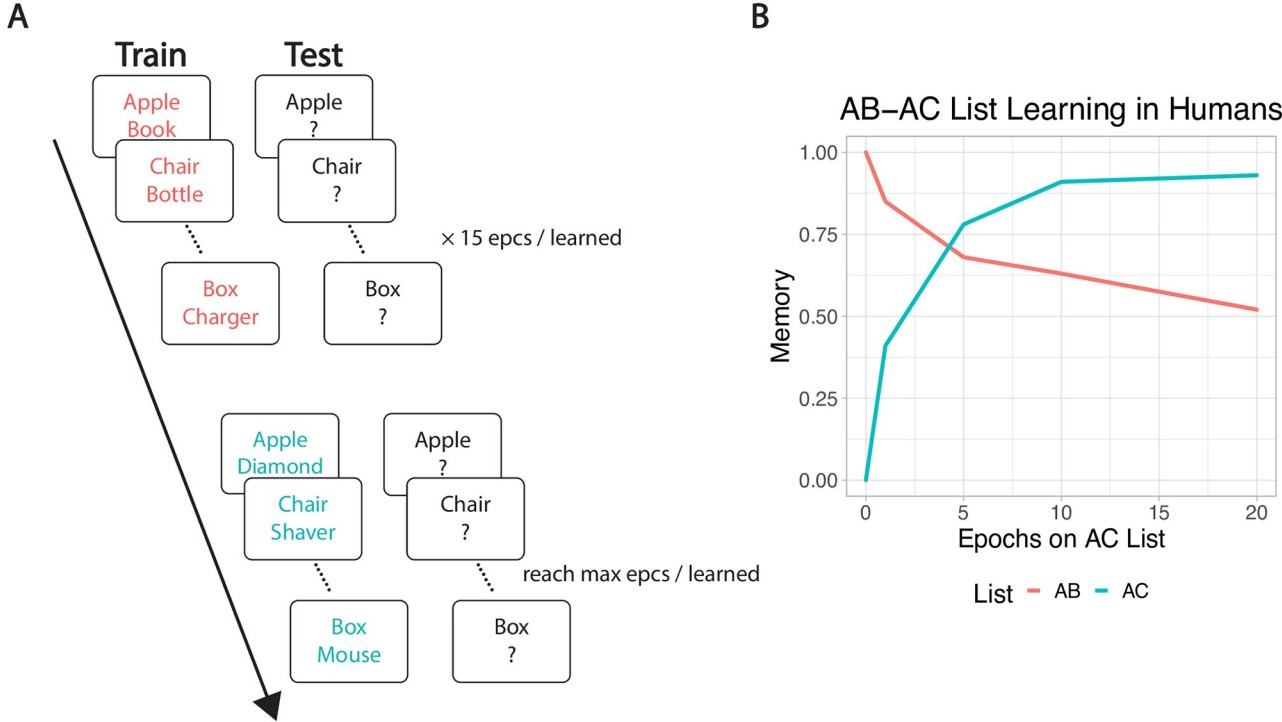

**Fig 2. AB–AC list learning paradigm diagram and human data reproduced from an empirical experiment [53]. A)** The first AB list is trained until memory accuracy reaches 100% or 15 epochs, whichever is less; the second AC list is then trained to same criterion, while continuing to test AB and AC items. Detailed procedure is described in Methods. **B)** Human participants show moderate interference of the AB list after learning the AC list.

## Model testing

The task used in the current study is a standard AB-AC paired-associates list-learning paradigm, widely used to stress interference effects [53, 54] (Fig 2). In these paradigms, typically, a participant learns a list of word pairs, with each pair referred to as *A-B*. Once the pairs are learned to a criterion or a fixed number of repetitions, participants learn a new list of *A-C* word pairs, in which the first word in each *A-B* pair is now associated with a new word. Learning of A-C pairs is typically slowed due to competition with previously learned A-B pairs (*proactive interference*), and once the A-C pairs are learned, retention of A-B pairs is reduced (*retroactive interference*).

To simulate the AB–AC paradigm, each pair of A and B items (unique random bit patterns in the model) was trained, and then tested by probing with the A item and testing for recall of the associated B item. A list context representation was also present during training and testing, to distinguish the AB vs. AC list (see S1 Appendix for an example pattern). Each pair was only trained once during one *trial* in an *epoch*, with each trial being a full theta cycle (~200 ms). All pairs (including AC pairs) were tested once in each epoch. For simplicity and compatibility with other settings in the Emergent software, we implemented the model using an alpha cycle (~100 ms), which does not make any functional difference, as confirmed by our testing. Once recall accuracy for all AB pairs reached 100%, or 15 epochs of the whole AB list have been trained, the model switched to learn the AC list, where previously learned A items were paired with novel C items and AC list context. Similarly, if memory for all AC pairs reached 100%, or 30 epochs have been trained in total, that run was considered complete. We ran 30

different simulated subjects (i.e., runs) on each configuration and set of parameters, with each subject having a different set of random initial synaptic weights.

There are several central questions that we address in turn. First, we compared the earlier theta-phase hippocampus model with the new Theremin model to determine the overall improvement resulting from the new error-driven learning mechanism and other optimized parameters. This provides an overall sense of the importance of these mechanisms for episodic memory performance, and an indication of what kinds of problems can now be solved using these models, at a practical level. In short, the Theremin model can be expected to perform quite well learning challenging, overlapping patterns, opening up significant new practical applications of the model.

Next, we tested different parameterizations of the Theremin model, to determine the specific contributions of: 1) adding error-driven learning in the CA3, compared to Hebbian learning in this pathway, with everything else the same (NoEDL variant); 2) reduced MF → CA3 strength during testing (cued recall, NoDynMF variant); 3) the balance of weight decreases vs. increases in the ECin → DG projections; 4) ECin → DG Hebbian learning, compared to no learning (NoDGLearn variant); 5) the effect of pretraining on the monosynaptic pathway between EC and CA1 (NoPretrain variant), which simulates the accumulated learning in CA1 about the semantics of EC representations, reflecting in turn the slower learning of cortical representations. In other words, human participants have extensive real life experience of knowing the A/B/C list items, enabling the CA1 to already be able to invertably reconstruct the corresponding EC patterns for them, and pretraining captures this prior learning. Pretraining has relatively moderate benefits for the Theremin model, and was used by default outside of this specific test. The pretraining process involved turning DG and CA3 off, while training the model with items and context separately only in the monosynaptic EC ↔ CA1 pathway for 5 epochs.

The learning capacity of a model is proportional to its size, so we tested a set of three network sizes (small, medium, large, see S1 Appendix for detailed parameters) to determine the relationship between size and capacity in each case. The list sizes ranged from 20 to 100 pairs of AB–AC associations (for comparison, [53] used 8 pairs of nonsense syllables). For the basic performance tests, the two dependent variables were *number of epochs, N* and *residual AB memory, M*. *N* measures the total number of epochs used to finish one full run through AB and AC lists, which measures the overall speed of learning (capped at 30 if the network failed to learn). *M* is the memory for AB pairs after learning the AC list, thus representing the models' ability to resist interference.

In addition to these performance tests, we ran representational analyses on different network layers (i.e., hippocampal subregions) over the course of learning. This enabled us to directly measure the temporal difference error signals that drove learning in Theremin, and how representations evolved through learning. Furthermore, by comparing across differences in learning algorithm and other parameters, we can directly understand the overall performance differences. The main analytic tool here is to compute cycle-by-cycle correlations between the activity patterns present at that cycle and the patterns present at the end of a trial of processing (100 cycles), which provides a simple 1-dimensional summary of the high-dimensional layer activation patterns as they evolve over time.

Finally, we ran a version of our model to simulate the testing effect in a behavioral experiment [55]. The testing effect is a widely-replicated finding that learning in the context of testing (e.g., with a partial retrieval cue to probe retrieval of previously-studied information, also known as *retrieval practice*) is more effective than re-studying the original complete information. It should be clear that this testing-based learning should activate greater error-driven

learning than restudy, and indeed we show that the Theremin model is better able to take advantage of these error signals than the comparison NoEDL model.

We used the small hippocampus with 100 pairs of AB (no AC in this task), and simulated the experiment with Theremin and NoEDL, each running 30 subjects (i.e., runs). Both models started with 5 epochs of pretraining and an epoch of initial learning all using the Theremin setting to achieve same initial criteria. After the initial learning, either retrieval practice (RP, i.e., testing) or restudy (RS) was run for another epoch, with a context drift of 10% to simulate the interval between learning and RP/RS in the experiment. Specifically, RP uses the same settings as testing in our model, but with the plus phase clamped to the correct answer (i.e., feedback in the experiment) for learning. The ECin → CA1 and ECout ↔ CA1 were turned off in RP to prevent learning in the empty item B pool, which would harm the learning of the models. Finally, an epoch of test was given, with another context drift of 10%, to test the final performance of both models.

## Results

### Overall memory performance

First, we examined the broadest measure of overall learning performance improvements in Theremin compared to the earlier theta-phase model from [21]. Fig 3 shows the results across all three network sizes and numbers of list items. For all three network sizes, the results show that Theremin was better at counteracting interference and retained more memory for AB pairs than the theta-phase hippocampus model across all list sizes and network sizes (Student's t-test, same for the following analyses, $p < .01$ except SmallHip List100 ($p = 0.736$)). Moreover, the full Theremin model completed learning significantly faster (i.e., the N measure) than the theta-phase hippocampus model across all list sizes and network sizes ($p < .01$ except SmallHip List100 (all N = 30)).

To more specifically test the effects of the new error-driven CA3 mechanism in the Theremin model, we directly compared the Theremin model with another Theremin model without error-driven CA3 component (labeled as NoEDL), but with everything else the same. For this and subsequent comparisons, we focus on the medium and large network sizes, as the small case often failed to learn at all for larger list sizes. Fig 4 shows that, except for the smallest list size (20 items), Theremin retained significantly more AB memory ($p < .01$, except large network with list size of 20 ($p = 0.321$)) and learned faster ($p < .01$, except large network with list size of 20 ($p = 0.343$)) than NoEDL. Thus, it is clear that this error-driven learning mechanism is responsible for a significant amount of the improved performance of the Theremin model relative to the earlier theta-phase model.

To determine the contributions of the other new mechanisms included in the Theremin model, we compared the full Theremin to versions without each of these mechanisms (Fig 4). The NoDynMF version eliminated the mechanism of dynamically decreasing the strength of MF inputs from DG to the CA3 during recall, and the results show a significant effect on performance for all but the smallest list size (20 items) (N $p < .01$, except large network with list size of 20 ($p = .013$), M $p < .01$, except large network with list size of 20 and 80 ($p = 0.127$)).

To determine the importance of learning in ECin → DG pathway overall, we tested a NoDGLearn variant with no learning at all in this pathway. In principle, the DG could support its pattern separation function without any learning at all, relying only on the high levels of pattern separation and random PP connectivity. However, we found that learning in this pathway is indeed important, with an overall decrease in performance for larger list sizes (above 40 items) (N $p < .01$, BigHip List40 $p = .011$; M $p < .01$, BigHip List40 $p = .025$). Interestingly, as the list size scaled up, the NoDGLearn model learned increasingly more slowly, such that it

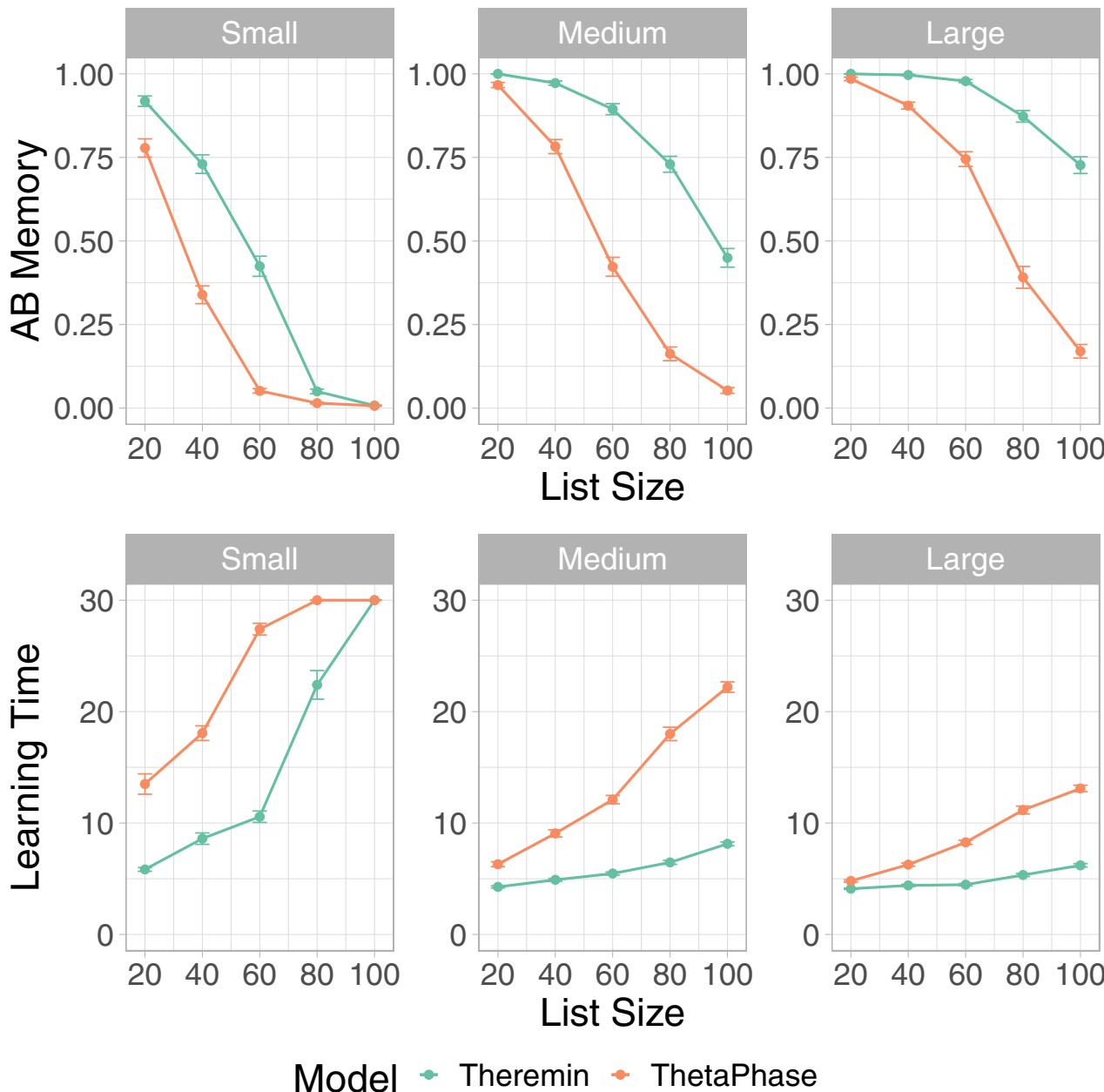

**Fig 3. Theremin vs. ThetaPhase on AB memory and learning time for all three network sizes.** The Theremin model was better at counteracting interference across all list sizes and network sizes, and had significantly faster training time across all list sizes and network sizes.

was even slower than the theta-phase model at a list size of 100. This effect is attributable to the strong effect of DG on training the CA3, and when the DG's ability to drive strong pattern separation is compromised, it significantly affects CA3 and thus the overall memory performance.

The higher rate of weight decrease (LTD = long-term depression in biological terms) relative to weight increases in the ECin → DG pathway were also important: eliminating this asymmetry significantly decreased performance for larger list sizes (above 60 items) (N $p <$ .01, SmallHip List60 $p$ = .051; M $p <$ .05, BigHip List80 $p$ = .129). We also found that a lower

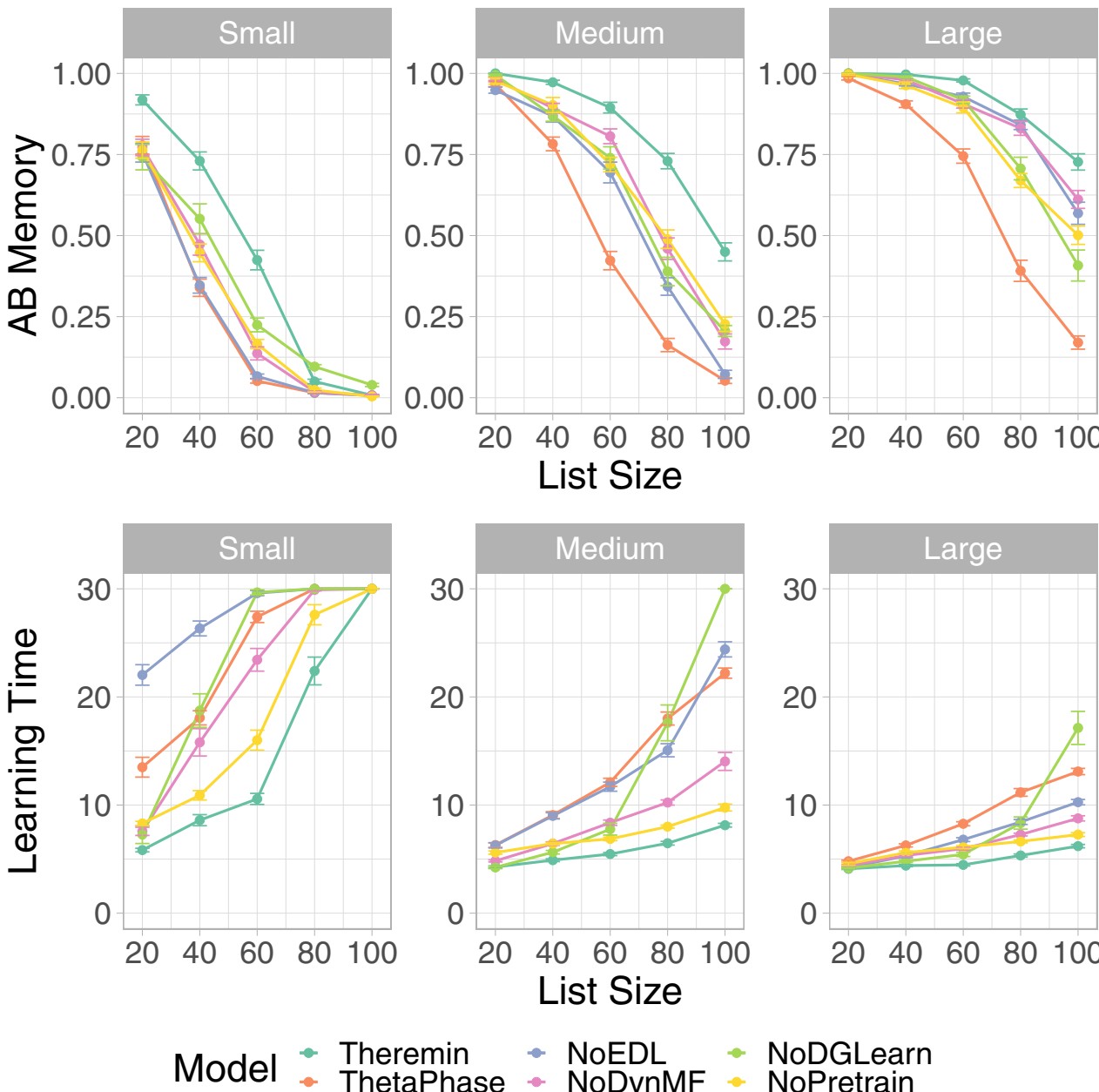

**Fig 4. Theremin vs. other models on AB memory and learning time.** Theremin and ThetaPhase data are the same as in Fig 3, shown for reference purpose here. NoEDL is the Theremin without the new error-driven learning mechanism. NoDynMF is the Theremin with same mossy fiber strength during training and testing. NoDGLearn is the Theremin with ECin → DG learning off. NoPretrain is the Theremin without pretraining CA1. Each of these factors makes a significant contribution as seen in decrements relative to the full Theremin in interference resistance (AB memory) and learning time.

learning rate in the ECin → DG pathway improved the M score (reducing interference), but resulted in slower learning, and vice-versa for higher learning rates, consistent with the fundamental tradeoff between learning rate and interference that underlies the complementary learning systems framework [5]. Likewise, due to optimized parameters in Theremin, comparing it to a lower or higher learning rate model would result in significant improvement in M

or N, respectively, but not both. Thus, we compared two Theremin variants that had dramatic differences in both M and N. Higher learning rate resulted in faster learning ($p < .01$) but less M ($p < .01$) compared to lower learning rate for list sizes over 40, vice versa.

The final mechanism we tested was the pretraining of the EC $\leftrightarrow$ CA1 encoder pathway, to reflect long-term semantic learning in this pathway. The NoPretrain variant showed significantly worse performance at all but the smallest list sizes (N $p < .01$; M $p < .05$ except BigHip List20 ($p = .155$)).

## Representational dynamics

Having established the basic memory performance effects of the error-driven CA3 and other mechanisms in the Theremin model, we now turn to some analyses of the network representations and dynamics to attempt to understand in greater detail how the error-driven learning shapes representations during learning, how the activation dynamics unfold over the course of the theta cycle within a single trial of learning, and how these dynamics change over multiple iterations of learning. For these analyses, we focus on the 100-item list size, and the medium sized network, comparing the full Theremin model vs. the NoEDL model, to focus specifically on the effects of error-driven learning in the CA3 pathways.

Fig 5 shows a representational similarity analysis (RSA) of the different hippocampal layers over the course of learning, comparing the average correlation of representations in CA3 within each list (all AB items and all AC items, e.g., A1B1 vs. A2B2) and between lists (AB vs. AC, e.g., A1B1 vs. A1C1). These plots also show the proportion of items correctly recalled from each list, with the switch over from the AB to AC list happening half-way through the run (we fixed this crossover point to enable consistent averaging across 30 simulated subjects, using a number of epochs that allowed successful learning for each condition). These results show that the error-driven learning in the full Theremin model immediately learns to decrease the similarity of representations within the list (e.g., WithinAB when learning AB) and between lists over training, while the Hebbian learning in the NoEDL model fails to separate these representations and results in increases in similarity over time. This explains the reduced interference and improved learning times for the error-driven learning mechanism, and is consistent with the idea that the continuous weight changes associated with Hebbian learning are deleterious.

Fig 6 shows an example AB pair plot, with each layer's correlation with the final activation state at the end of the trial across 4 training epochs. As illustrated in the plot, the learning dynamics in DG, CA3 and CA1 layers follow different learning rules across 4 quarters in one trial. In the CA3, error-driven learning in the Theremin model causes its activation to converge over the course of learning based on the target DG input that arrives starting after cycle 25. This learning progression is not evident in the NoEDL model, where Hebbian learning in the CA3 establishes a relatively stable representation early on. The CA1 shows increasing convergence to the final plus-phase pattern starting in the second and third quarter (cycle 26–75), when CA3 starts to drive CA1. Interestingly, there is evidence for a "big-loop" error signal [56] reflecting activation circulating through the trisynaptic pathway and out through the EC, and back into the CA3, deviating its pattern from the stabilized one, as depicted by the slightly curved green line in the first epoch.

To elaborate on the error-driven learning dynamics in the Theremin model, as learning progresses, the CA3 pattern in the first quarter becomes increasingly similar to its final pattern (Fig 6). In effect, this similarity signal reflects how close the CA3 pattern is to its final DG-dominated pattern, before DG starts to have an effect on CA3. In the first epoch, CA3 is driven only by ECin $\rightarrow$ CA3 and CA3 $\rightarrow$ CA3 inputs, resulting in a large temporal difference (error

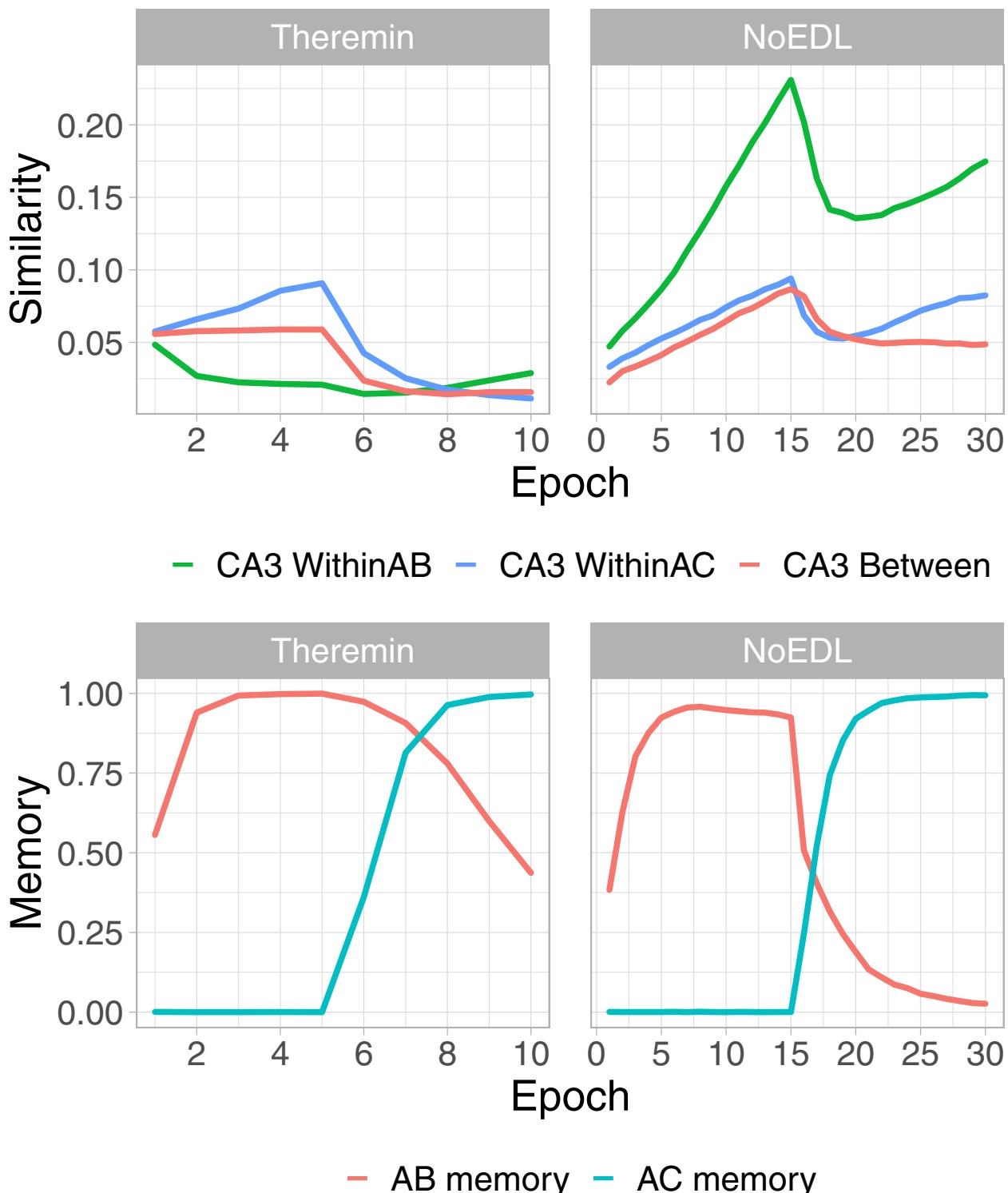

**Fig 5. Statistics for area CA3 over the course of testing (List 100, Medium sized network).** Representational similarity analyses (RSA) for area CA3 for Theremin vs. NoEDL show how the error-driven learning in Theremin reduces the representational overlap (top left) whereas the Hebbian learning in NoEDL increases the representational overlap (top right). This explains the differential interference as shown in the AB Memory plot for each case (bottom row). The number of epochs used in Theremin training was set to a fixed number (i.e., 10) that enabled complete learning of AB and AC lists, while in NoEDL was set to the maximum amount used in the current paper (i.e., 30).

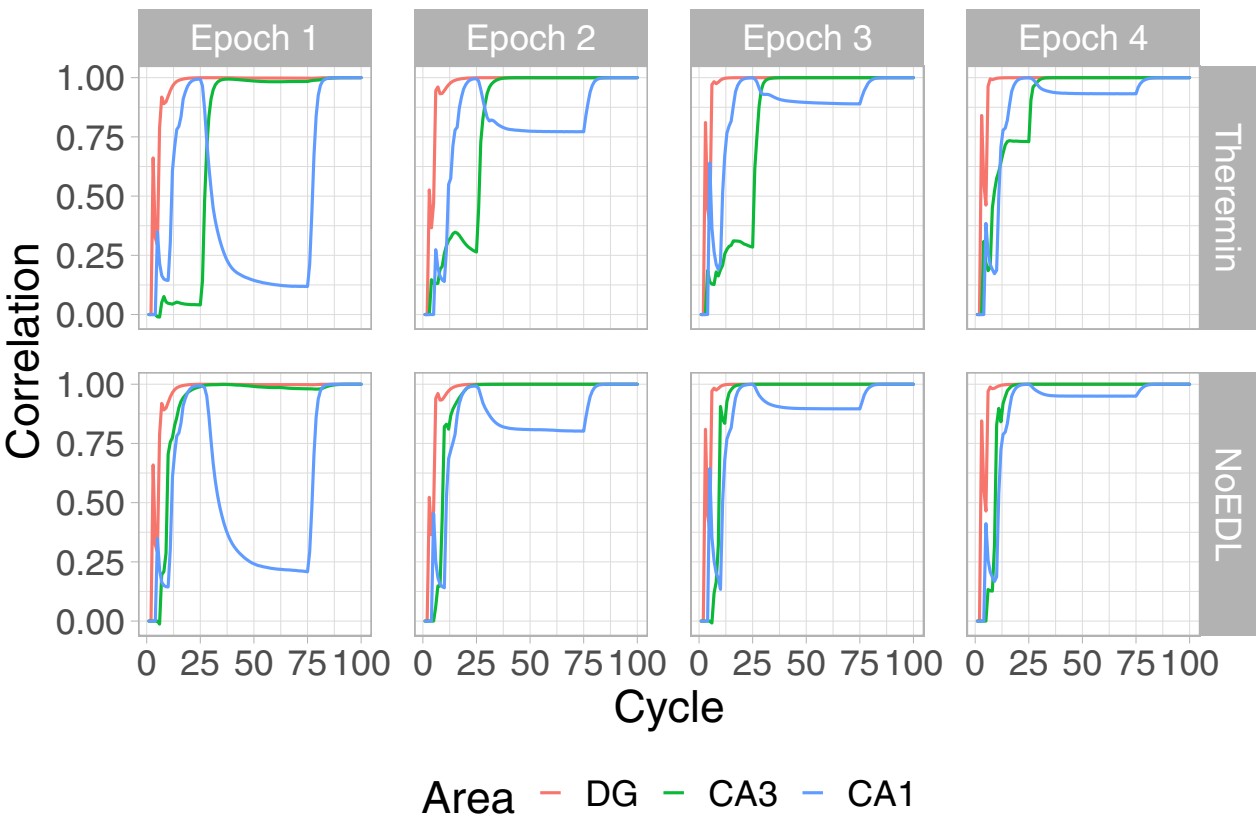

**Fig 6. Pattern similarity development throughout learning.** Changes in hippocampal subregions' pattern similarities over the course of the first 4 epochs of learning, within a full trial for an example AB pair (model timing equivalent to 100 ms), for the Theremin (top row) and NoEDL models (bottom row). Each line reflects the correlation of the current-time activity pattern relative to the activity pattern at the end of the trial. Two major effects are evident. First, the CA1 pattern learns over epochs to quickly converge on the final plus-phase activation state, based on learning in the CA3 → CA1 pathway. Second, the Theremin model shows how the CA3 pattern learns over epochs to converge on the DG-driven activation state that arises after cycle 25, reflecting CA3 error-driven learning. Additionally, big-loop signals from ECout back to ECin could be observed from cycle 25 to 75 in the first epoch for both models (shifting CA3 patterns slightly off its final patterns). Drops seen within the first few cycles were due to the settling of temporally different patterns and were not of interest to the current paper.

signal), which in turn modifies these connections (i.e., heterosynaptic plasticity). This error becomes smaller fast, and learning will stop when there is no more error. On the other hand, the NoEDL model continually increases the synaptic weights between CA3 and other regions whenever two neurons are active together, according to the Hebbian learning principle.

### Testing effect by error-driven learning hippocampus

Finally, we show that the testing effect could arise due to the error-driven learning dynamics in the hippocampal CA3. Fig 7 shows that retrieval practice (RP) produced significantly better memory than restudy (RS) in the Theremin ($p < .01$), consistent with the behavioral findings. However, testing without the error-driven learning dynamics resulted in catastrophic interference, causing a significant negative testing effect in the NoEDL ($p < .01$). This suggests that the CA3 error-driven dynamics in the Theremin actually benefits from larger errors created by the process of retrieval and the feedback in the RP, compared with another epoch of training in the RS.

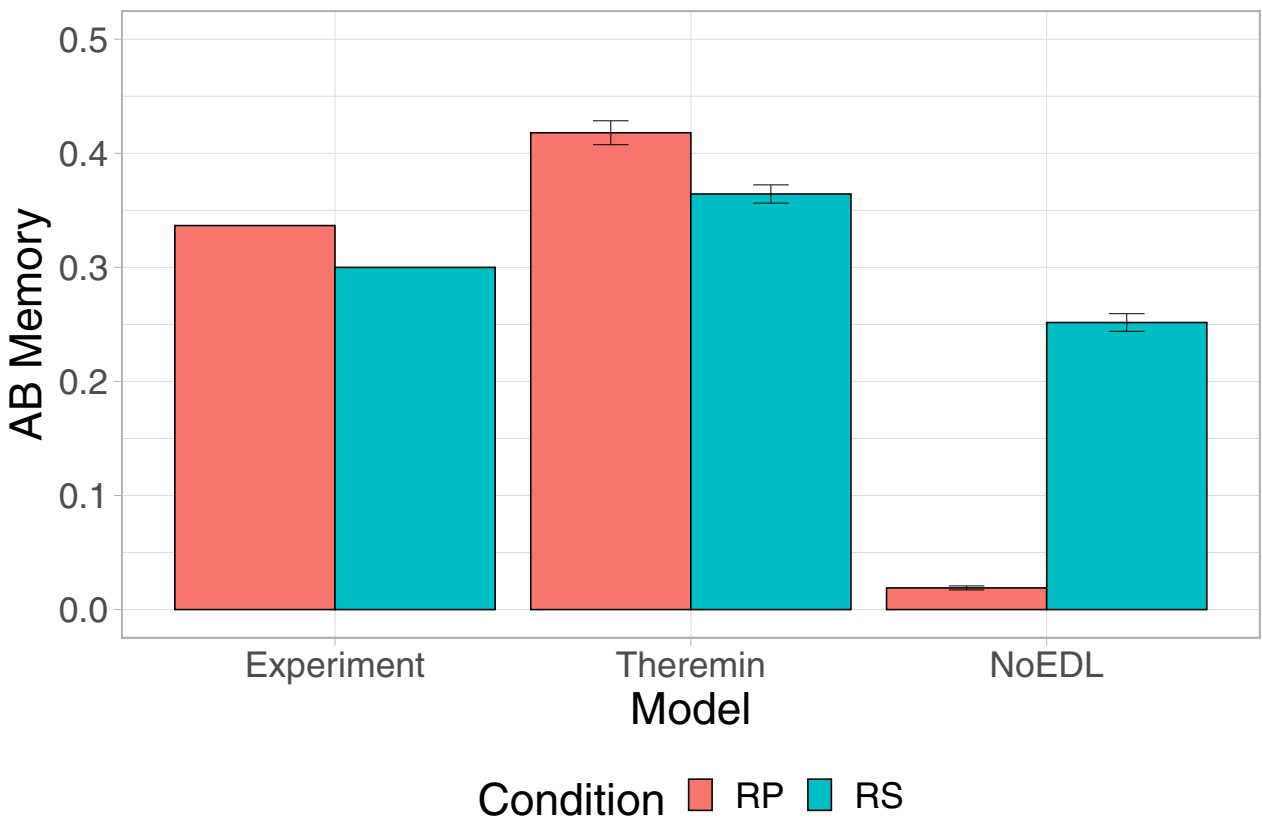

**Fig 7. Testing effect simulation.** The testing effect refers to memory increase when retrieving learned information, compared to restudying. In the experiment [55], subjects learned 30 Eskimo/English word pairs, went through either retrieval practice (RP) or restudy (RS), and then got a final test. We modeled a similar process using the Theremin model and the NoEDL model (for modeling details, see Methods). Results show that the Theremin model was able to achieve qualitative fit to the experiment data, while the NoEDL model lost desirable memory performance in RP compared to RS, suggesting that error-driven learning might be crucial to the testing effect.

## Discussion

By incorporating biologically plausible error-driven learning mechanisms at the core CA3 synapses in our computational model of the hippocampus, along with a few other important optimizations, we have been able to significantly improve learning speed and memory capacity (resistance to interference) compared to our previous model that used Hebbian learning in these synapses. These results demonstrate the critical ability of error-driven learning to automatically limit further learning once it has achieved sufficient changes to support effective memory recall, which then significantly reduces the amount of interference that would otherwise occur from continued synaptic changes. Furthermore, representational similarity analysis (RSA) was used to illustrate temporal dynamics within the hippocampal formation, which explains the effects of the error-driven learning mechanism, making it possible for the model to make specific subregional predictions that could be tested in experiments. Finally, by simulating the testing effect, we showed that the error-driven dynamics in the hippocampal CA3 could be critical to this long-standing behavioral phenomenon, and provide a new perspective in terms of neural computation underlying this effect.

Although we have provocatively characterized Hebbian learning as a mistake in order to highlight the error-correction nature of our alternative hypothesis, we nevertheless recognize

that extensive research and modeling has productively leveraged the Hebbian principle to understand hippocampal function. Indeed, we have shown that the error-driven learning achieves much of the same overall learning objective, just through different means that improve learning performance and reduce interference. As emphasized above, this interference is not a result of using a simplistic form of Hebbian learning without appropriate normalization and bounding—the key failing of Hebbian learning that error-driven learning corrects is that it is purely local and autonomous and is not sensitive to an overall objective function that can determine when learning has accomplished its objective. In addition, there are various other lines of research about non-Hebbian learning in the hippocampus that should be acknowledged, and future work can investigate further [40, 42, 57–59].

Another contribution of the current model is to test several computationally-motivated mechanisms that improve overall performance, and could plausibly be implemented in the hippocampal biology. First, decreasing the DG → CA3 strength during recall improves performance, because the DG otherwise biases more strongly toward pattern separation rather than the pattern completion needed for recall [6, 60]. In addition, entirely eliminating the MF projections during testing actually harms memory recall. These findings are consistent with data and models showing that MF projections are not necessarily needed during recall [44, 46, 47], but may help increasing recall precision [44, 46, 48, 49]. Consistent with this perspective and the contribution of DG to recall performance (going beyond its widely-discussed pattern-separation contributions), we found that learning in the ECin → DG pathway is important for overall performance. Furthermore, favoring of LTD over long-term potentiation (LTP) in this pathway is beneficial as it forces DG to form sparse representations, suggesting that learning overall is helping with the DG pattern separation dynamics. This echoes with the idea that homeostatic synaptic plasticity at this pathway helps convey the most favored patterns [61].

Finally, by doing pretraining, prior semantic knowledge in the CA1 area benefits subsequent learning and memory. Although our previous model without all of these improvements was sufficient for simulating smaller-scale one-off experiments, the significantly improved capacity of the present model opens up the potential to examine longer time-scale learning contributions of the hippocampus, and other larger-scale datasets as emphasized in [25].

Overall, the Theremin model retains the major tenets of the Hebb-Marr paradigm based on rapid episodic learning, while incorporating error-driven learning to optimize the learning capacity of the system relative to the predominant use of Hebbian learning in other models. In the following subsections, we consider other theoretical models of the hippocampus that can usefully be compared with the present model, including a number of widely-cited theories that postulate some form of error-signaling or error-driven learning. At the heart of many of these models is the idea that the hippocampus can generate predictions in order to then compute a novelty or error signal relative to such predictions, or to learn and predict sequences of future states. After briefly summarizing these models, we discuss what roles the hippocampus and the neocortex play in generating predictions according to the complementary learning systems (CLS) framework in which the current model is based [5, 9, 62].

## Prediction-based models of the hippocampus

One longstanding and influential set of theories suggests that the hippocampus acts as a *comparator*, generating predictions in order to detect and signal *novelty* or *surprise* [63–65]. Specifically, the hippocampus in these models generates a global *scalar* signal as a function of the relative mismatch between a predicted state and the actual next state. For example [63] proposed that combining previous sensory information and the motor plan creates predictions about the current state, which are then compared with the actual current sensory information.

The motor plan is maintained if the two states match, but it is interrupted in the case of incorrect predictions (i.e., surprise or novelty), so that the animal can attempt to solve the problem in a different way. In the [65] model, the hippocampal novelty or surprise signal is hypothesized to drive phasic dopamine firing via its subcortical projection pathway through the subiculum. These different models vary in terms of the exact mechanisms and subfields proposed to compute the mismatch signal (CA3, CA1 or subiculum), but they assume a similar overall functional role for the hippocampus in terms of synthesizing predictions.

In the present model, error signals are not summed, but rather used to optimize learning of specific associations. However, it is possible that the temporal-difference error signals present in our hippocampal model could play a role in generating a global novelty signal. For example, at different points in the theta phase cycle (Fig 1), area CA1 and ECout are representing the current information as encoded in CA3 and its projections into CA1, versus the bottom-up cortical state present in ECin. The difference between these two activation states could be converted into a global mismatch signal that would reflect the relative novelty of the current state compared to prior episodic memory learning in the CA3 of the hippocampus. Likewise, it is possible that a similar global error signal could be computed from the temporal differences over CA3 in our model, reflecting the extent to which CA3 has learned to encode the more pattern-separated DG-driven pattern, which is likely to also reflect the relative novelty of the current input state. We will investigate these possibilities in future research.

Prediction-based learning in the hippocampus is also central to another early computational model, which is based on error-driven backpropagation learning in the context of a predictive autoencoder [66]. In this model, the hippocampal network learns by simultaneously attempting to recreate the current input patterns, and also predict future reinforcement outcomes. The cortical network representations are then shaped by hippocampal training signals, similar in spirit to the scalar novelty / surprise signals. Simulations with this model and its hippocampus-lesioned variant have been shown to replicate a wide range of conditioned behaviors in rats and rabbits [67], although it is notable that many of these same phenomena can also be accounted for using an earlier version of the episodic memory model presented here [51].

Another class of models hypothesizes that the hippocampus learns *sequences* of events over time, such that, when a past state is encountered, the hippocampus can enable the prediction of potential outcomes of actions taken in novel situations, based on what has happened previously [24, 47, 68–72]. In some of these models, the recurrent connections in area CA3 learn to associate prior time step representations with subsequent time step patterns, thus learning to predict the next state based on the current state. Other models suggest that the hippocampus learns systematic predictive representations (e.g., a successor map of subsequent states following the current state in the case of [72]). Most of the models suggest that the hippocampus itself is capable of synthesizing novel predictions based on these learned sequences.

## Neural mechanisms of prediction in hippocampus and cortex

The models discussed above emphasize the idea that memory retrieval in the hippocampus is a form of prediction, and at a broader level, many researchers have embraced the idea that the hippocampus might be specialized for generating predictions in the service of navigation, reasoning, and imagination [73–78, 78–80]. These theories, however, tend to describe prediction in broad strokes, and as such, we argue that they do not respect the computational limitations of the hippocampus.

In contrast to the above models, we do not believe that the hippocampus itself is well-suited for generating predictions in novel situations, and instead we think the relevant data can be

better captured in terms of the simple episodic memory framework that the Hebb-Marr model embodies (as updated in the present paper). Here, the hippocampus is specialized for rapidly encoding memories of distinct events or episodes using highly pattern-separated representations, which can later be recalled through the process of pattern completion. Given the overwhelming empirical support for the idea that the hippocampus is specialized for rapidly learning new episodic memories, we believe that it also cannot support a semantic prediction system capable of generating systematic predictions in novel situations.

Specifically, generating a novel prediction typically requires a cognitive process to synthesize prior experience and general principles (e.g., a scientific theory, or implicitly-learned regularities of the world, such as intuitive physics) to specify what will happen in the future. This kind of systematic generalization from prior experience to novel situations is precisely what the neocortex is thought to be optimized for according to the CLS theory [5, 9, 62]. This is because the overlapping representations of cortical networks are optimized to slowly integrate statistical regularities across many different experiences to learn *semantic* representations capable of supporting systematic generalization in novel situations. Indeed there are various models of error-driven predictive learning in the neocortex capable of learning such systematic predictive abilities, including a biologically-detailed proposal based on thalamocortical loops [36].

Although the computational architecture of the hippocampus is not well-suited for generating predictions on its own, it can certainly provide relevant episodic memories as input to the cortical prediction generation process. For example, strategic recall of particular memories, followed by appropriate updating of the details to better match the current circumstances, could produce a more generative predictive system that can synthesize novel predictions for new situations. These kinds of complex interactions, however, go well beyond the capabilities of the hippocampal circuit by itself, as captured in any implemented computational model.

## Nonmonotonic plasticity vs. error-driven learning

The temporal difference error signals that drive learning in our model can be related to the neural activation signals that drive nonmonotonic plasticity (NMP) learning dynamics as explored by Norman and colleagues [81]. Specifically, the nonmonotonic plasticity function drives LTD when activations are at a middling, above-zero level, while LTP occurs for more strongly activated neurons. This is the same underlying learning function that we use in our error-driven learning model [35], and thus it can be difficult to strongly distinguish the predictions of these two models. In particular, the conditions under which errors drive LTD in our model can be construed as being within the LTD range of the nonmonotonic plasticity function, under various additional assumptions. However, the NMP models have not been implemented within the context of a full hippocampal circuit, and it is unclear how those models might actually perform in specific conditions.

## Novel predictions

There are several novel, testable predictions from our model that can distinguish it from a more Hebbian-based model:

- As shown in Fig 5, the error-driven learning in area CA3 serves to drive pattern separation over time among otherwise similar representations, whereas the Hebbian version of the model showed increasing patterns similarity over learning. Thus, experiments that track the progression of representational similarity over the course of learning could distinguish these two patterns.

- By experimentally canceling the temporal difference of DG → CA3 and ECin → CA3, learning might still be preserved but impaired to a large extent. Similarly, CA1 error-driven learning [21] depends critically on the modulation of different pathways of connectivity within the hippocampus, organized according to the theta cycle in rodents according to [22], creating the temporal differences that drive CA1 learning. It was proposed that CA3 might also have encoding and retrieval modes at troughs and peaks of a theta cycle [60], but it is unclear how such model would benefit CA3 learning without an explicit role of DG, which was lacking in the only one experimental confirmation of this model [82]. Thus, neural manipulations that selectively disrupt the theta cycle and / or these pathway-specific modulations should disrupt error-driven learning, therefore decreasing learning ability, but may not affect recall of previously-learned information to the same extent. By contrast, it is not clear why from the purely Hebbian learning framework that disrupting the theta cycle should impair that form of learning. Intriguingly, a recent report appears to be consistent with the predictions of our model: [83] found that a highly selective disruption of the timing of the theta cycle produced selective deficits in learning, but not retrieval.

- Our model also generates novel predictions about the functional characteristics of human memory. For instance, there is a large body of evidence about the *testing effect*, in which items that are tested with partial information (as compared to restudy of the complete original information) are better retained than items that are re-studied [26]. The superiority of testing over restudy presents a challenge to models depending on Hebbian learning because learning a precise input pattern should be as good or better than learning from a partial cue. Theremin, however, provides a natural explanation for the testing effect, as the difference between an initial guess and subsequent correct answer provides an opportunity for error-driven learning, whereas restudy provides no opportunity to make the initial guess needed in order to optimize weights. We illustrated this idea in a concrete simulation and suggest that error-driven learning might be a key component to the underlying neural computation of testing effect.

## Conclusions

In summary, results from our simulations show that error-driven learning mechanisms can dramatically improves both memory capacity and learning speed by reducing competition between learned representations. Furthermore, these mechanisms can potentially explain a wide range of learning and memory phenomena. Error-driven learning in CA3 can emerge naturally out of neurophysiological properties of the hippocampal circuits, building on the basic framework for error-driven learning in the monosynaptic EC ↔ CA1 pathway [21]. There are many further implications and applications of this work, and many important empirical tests needed to more fully establish its validity. Hopefully, the results presented here provide sufficient motivation to undertake this important future research.

## Supporting information

**S1 Appendix. Appendix text.**
(PDF)

**S1 Fig. Training and testing example patterns for network input.** Each input pattern has 6 pools, composed of 2 item pools at the bottom (i.e., A at bottom left; B or C at bottom right) and 4 list context pools at the middle and the top rows—memory of AB and AC pairs are categorized into different experiences, with four different context pools for each experience. During testing, pool A and four context pools were presented as in training, while pool B/C at the

bottom right was silent, therefore challenging the model to retrieve a memorized answer at the bottom right pool of ECout.
(EPS)

**S1 Table. Parameters for network sizes.** In neural networks, larger network size usually leads to higher capacity, when controlled for other settings. In the current study, we tested different variations of the hippocampus model for three different network sizes to show the benefit of error-driven learning for hippocampus regardless of sizes, meaning the mechanism is generalizable. For pool sizes, the numbers in the table refer to number of neurons in that specific pool. Note: DG size is around five times CA3 size as specified in our previous model [21].
(PDF)

**S2 Table. Parameters.** Non-default parameters used in the model, with default shown.
(PDF)

# Author Contributions

**Conceptualization:** Yicong Zheng, Satoru Nishiyama, Charan Ranganath, Randall C. O'Reilly.

**Funding acquisition:** Charan Ranganath, Randall C. O'Reilly.

**Investigation:** Yicong Zheng, Xiaonan L. Liu, Satoru Nishiyama, Randall C. O'Reilly.

**Methodology:** Yicong Zheng, Xiaonan L. Liu, Satoru Nishiyama, Charan Ranganath, Randall C. O'Reilly.

**Project administration:** Randall C. O'Reilly.

**Resources:** Charan Ranganath, Randall C. O'Reilly.

**Software:** Yicong Zheng, Xiaonan L. Liu, Satoru Nishiyama, Randall C. O'Reilly.

**Supervision:** Xiaonan L. Liu, Charan Ranganath, Randall C. O'Reilly.

**Validation:** Yicong Zheng, Randall C. O'Reilly.

**Visualization:** Yicong Zheng, Randall C. O'Reilly.

**Writing – original draft:** Yicong Zheng, Charan Ranganath, Randall C. O'Reilly.

**Writing – review & editing:** Yicong Zheng, Xiaonan L. Liu, Satoru Nishiyama, Charan Ranganath, Randall C. O'Reilly.

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
