## [Decision Letter · Decision Letter 0]

24 Jan 2022

Dear Dr. O'Reilly,

Thank you very much for submitting your manuscript "Correcting the Hebbian Mistake: Toward a Fully Error-Driven Hippocampus" for consideration at PLOS Computational Biology.

As with all papers reviewed by the journal, your manuscript was reviewed by members of the editorial board and by several independent reviewers. In light of the reviews (below this email), we would like to invite the resubmission of a significantly-revised version that takes into account the reviewers' comments. More specifically, as mentioned by Reviewer #1, it is important if you could show that the proposed model can capture a relevant phenomenon in addition to improving performance.

We cannot make any decision about publication until we have seen the revised manuscript and your detailed response to the reviewers' comments. Your revised manuscript will be sent to reviewers for further evaluation.

Sincerely,

Alireza Soltani

Associate Editor

PLOS Computational Biology

Kim Blackwell

Deputy Editor

PLOS Computational Biology

Reviewer's Responses to Questions

**Comments to the Authors:**

Reviewer #1: Review of PLoS Comp Biol article Dec 2021

The paper shows how an alteration to connection-strength update rules in a multi-region connectionist model of hippocampal function leads to improved performance—in terms of faster learning and less interference—in associative learning tasks. The results are interesting as are the updates to the learning rule. My main concern is that the authors assume the reader will know or read a whole series of prior papers presenting models upon which this one is built – the methods presented here are simply inadequate for a reader to fully understand, yet alone reproduce the work. The authors send the reader to a githib repository, but even there basic features are unclear – for example, whether “Adaptive Exponential Leaky Integrate and Fire” model neurons are used, or firing-rate models based on the latter. These are the basic building blocks of any model, along with the impacts of connections – amplitudes and time constants, most importantly. The reader should not have to wade through all the prior papers to figure out what procedures might have been used in this one. I suggest providing full tables of the form of Nordlie et al, PLoS Computational Biol 2009.

My second major thought is that to make small adaptations to a prior model in order to improve performance in a single task is not really newsworthy to any others than those who work on this somewhat niche sets of models. I think at least it should be shown that a second, more interesting phenomenon, such as the “testing effect” with partial information, which the authors say could arise from their model, should be validated. In this reviewer’s mind, that would make the paper stronger and a bit more general.

Some examples of the need for better explanation follow:

The equations shown are not mathematically rigorous. For example, equations 3 and 4 which indicate the rule used to update specific connection weights should indicate the two units (e.g. i and j) stating which is presynaptic, which is postsynaptic, and then how it depends on the firing rates r of either unit i or unit j.

Equation 3 is particularly confusing as the difference in rates used is of vectors of different sets of units in entrorhinal cortex (layer 3 versus deep layers) which would not be the same size biologically speaking. Since it is impossible to take the difference of two vectors of different lengths, the authors must assuming identical numbers of units and then a one-to-one correspondence between units in deep layers with equivalent units in layer 3. Such important details and clear constraints/biological requirements for a model should be stated plainly so they can be tested.

Since Equation 4 is the new one being used for this paper, again it would be better to see it written more accurately in terms of the time of firing rates. That is, is dW_ij for a connection from unit i to unit j equal to the firing rate of unit i in EC to the rate of unit j in CA3 at time t¬-tau subtracted from its rate at time t for a value of tau equal to a quarter of a theta cycle, so about 30ms? If this is the case (my best guess given what is presented) then it is essential that it is stated plainly and either evidence for such a plasticity rule provided, or a clear statement that there must be a process which responds to the *change* in firing rate over a 30ms period (rather than the absolute rate) so that ideally experimental groups could look for it and find the corresponding biochemical process, or those who model synaptic plasticity could suggest a mechanism for it, using known processes. Without such clear statements of the requirements for a model showing how it can be disproven, there is little benefit in adding alteration to alteration of a complicated model that may or may not correspond to the underlying biology. I see there are suggestions, perhaps based on work by Hasselmo’s group on different signs of plasticity at different phases in the theta cycle, but the exact requirements and equations are not provided. The text at the top of p.8 (which is far from Figure 1 on p.4, may need to be expanded and connected better with that figure’s caption.

On p.9 the authors state they use in their default setting, a pretraining process that involved “turning DG and CA3 off”. While it is reasonable to assume there are cortical representations of words in our vocabulary, a bit more justification is needed, given its limited capacity, and the ease of plasticity and interference in hippocampal areas, to (1) state evidence for long-term hippocampal representations of vocabulary and (2) justification for essentially switching off hippocampal structures while such representations are acquired. Perhaps (2) is to ensure cortical representations arise (in EC) without (1) in HC, but if that is the case more explanation is needed, as it looks like EC to CA1 synapses are “trained” which would result in CA1 representations of long-term semantics.

Minor:

The term “epoch” is not defined in the text or methods. Fig. 6 suggests 100 cycles per epoch? Is a cycle a theta cycle, about 100ms, so epochs are 10 sec long?

p.16 “the difficulties are more at the level of abstract principles” – this statement is confusing, as it seems the models are so different at the biological level the “difficulties” are far from abstract. It is pretty easy to replace a plasticity rule in an architecture and test its consequences, or to find out what parameters are needed for the rule to work – nothing is abstract about that.

The authors use in italics “ps” a lot when they I think mean p-values in significance tests? This is non-standard – I think just “p” is fine, but also state what sort of test is used.

Reviewer #2: My review is uploaded as an attachment.

Reviewer #3: Attached.

**Have the authors made all data and (if applicable) computational code underlying the findings in their manuscript fully available?**

Reviewer #1: Yes

Reviewer #2: Yes

Reviewer #3: Yes

PLOS authors have the option to publish the peer review history of their article (what does this mean?). If published, this will include your full peer review and any attached files.

Reviewer #1: No

Reviewer #2: No

Reviewer #3: No
---

## [Decision Letter · Decision Letter 1]

24 Aug 2022

Dear O'Reilly,

Thank you very much for submitting your manuscript "Correcting the Hebbian Mistake: Toward a Fully Error-Driven Hippocampus" for consideration at PLOS Computational Biology. As with all papers reviewed by the journal, your manuscript was reviewed by members of the editorial board and by several independent reviewers. The reviewers appreciated the attention to an important topic. Based on the reviews, we are likely to accept this manuscript for publication, providing that you modify the manuscript according to the review recommendations. 

**More specifically, there are minor suggestions by both Reviewer # 1 and 3 that should be considered. In addition, it seems that Reviewer # 2's concerns about implementing additional forms of self-limiting learning rules to compare with the performance of error-driven learning was not fully addressed. Because of this, the claims of the manuscript seem too strong in some places and could be toned down.**

Please prepare and submit your revised manuscript within 30 days. If you anticipate any delay, please let us know the expected resubmission date by replying to this email. **To avoid multiple round of reviews, your revised manuscript will be evaluated at the editorial level.**

Sincerely,

Alireza Soltani

Academic Editor

PLOS Computational Biology

Kim Blackwell

Section Editor

PLOS Computational Biology

[LINK]

Reviewer's Responses to Questions

**Comments to the Authors:**

Reviewer #1: It looks like prior critiques have been addressed well. A couple of minor suggestions:

Eq.2: When using the sum notation it is standard to include the subscripts (W_ij, x_j then sum over j) otherwise it could simply be written as a matrix multiplying a vector without an explicit summation.

P.4 "how testing effect" I think "how the testing effect"

Fig 7 caption "gained undesirable memory performance in RP compared to RS" : I think you mean the opposite. At least RS has high performance compared to RP in the right-hand columns. Maybe you mean it "lost desirable memory performance" which has a slightly different meaning as what is "desired" is to match the data. i.e. there is no gain in performance in RP.

Fig.8: I think it needs a bit more labeling in the figure, which pools are items etc, and more in the caption why some blocks are silent during the "test" , which blocks represent A or B or C etc.

Reviewer #3: The authors have clarified several areas of confusion adequately. I maintain that they can improve their scholarship in some areas by actually providing details of the previous literature they cite. For example, the authors write "building on theta-phase dynamics discovered by Hasselmo, Bodelon, and Wyble (2002)," but do not elaborate on the specific theta-phase dynamics they will be leveraging (although readers can probably figure this out by pulling up the citation and reading the paper). In another example, the authors write "Second, this error-driven learning could arise from heterosynaptic plasticity in the hippocampus (Lee, 2022)." The authors should state what potential heterosynaptic mechanisms exist (e.g. evidence of heterosynaptic plasticity by which neuromodulators at different phases of the theta cycle) and provide citations. The paper they cite is a model comparing successor representation learning algorithms to the "heterosynaptic plasticity rule." I think it would improve readability for the authors to go through the manuscript and determine if there are places where a citation is provided instead of important details + citations.

**Have the authors made all data and (if applicable) computational code underlying the findings in their manuscript fully available?**

Reviewer #1: Yes

Reviewer #3: Yes

PLOS authors have the option to publish the peer review history of their article (what does this mean?). If published, this will include your full peer review and any attached files.

Reviewer #1: No

Reviewer #3: No

Figure Files:

Data Requirements:

Reproducibility:

References:

---

## [Editor Report · Decision Letter 2]

19 Sep 2022

Dear O'Reilly,

We are pleased to inform you that your manuscript 'Correcting the Hebbian Mistake: Toward a Fully Error-Driven Hippocampus' has been provisionally accepted for publication in PLOS Computational Biology.

Best regards,

Alireza Soltani

Academic Editor

PLOS Computational Biology

Kim Blackwell

Section Editor

PLOS Computational Biology

---

## [Editor Report · Acceptance letter]

29 Sep 2022

PCOMPBIOL-D-21-02014R2 

Correcting the Hebbian Mistake: Toward a Fully Error-Driven Hippocampus

Dear Dr O'Reilly,

I am pleased to inform you that your manuscript has been formally accepted for publication in PLOS Computational Biology. Your manuscript is now with our production department and you will be notified of the publication date in due course.

With kind regards,

Agnes Pap
